# Phthalic Acid Esters: Natural Sources and Biological Activities

**DOI:** 10.3390/toxins13070495

**Published:** 2021-07-16

**Authors:** Ling Huang, Xunzhi Zhu, Shixing Zhou, Zhenrui Cheng, Kai Shi, Chi Zhang, Hua Shao

**Affiliations:** 1State Key Laboratory of Desert and Oasis Ecology, Xinjiang Institute of Ecology and Geography, Chinese Academy of Sciences, Urumqi 830011, China; huangling201@mails.ucas.ac.cn (L.H.); zhoushixing16@mails.ucas.ac.cn (S.Z.); czr24czr@163.com (Z.C.); shikai19@mails.ucas.ac.cn (K.S.); 2Research Center for Ecology and Environment of Central Asia, Xinjiang Institute of Ecology and Geography, Chinese Academy of Sciences, Urumqi 830011, China; 3Institute of Botany, Jiangsu Province and Chinese Academy of Sciences, Nanjing 210014, China; zhuxunzhi@cnbg.net; 4University of Chinese Academy of Sciences, Beijing 100049, China; 5Shandong Provincial Key Laboratory of Water and Soil Conservation and Environmental Protection, College of Resources and Environment, Linyi University, Linyi 276000, China

**Keywords:** phthalic acid esters, natural sources, biological activity, di-*n*-butyl phthalate, di(2-ethylhexyl) phthalate

## Abstract

Phthalic acid esters (PAEs) are a class of lipophilic chemicals widely used as plasticizers and additives to improve various products’ mechanical extensibility and flexibility. At present, synthesized PAEs, which are considered to cause potential hazards to ecosystem functioning and public health, have been easily detected in the atmosphere, water, soil, and sediments; PAEs are also frequently discovered in plant and microorganism sources, suggesting the possibility that they might be biosynthesized in nature. In this review, we summarize that PAEs have not only been identified in the organic solvent extracts, root exudates, and essential oils of a large number of different plant species, but also isolated and purified from various algae, bacteria, and fungi. Dominant PAEs identified from natural sources generally include di-*n*-butyl phthalate, diethyl phthalate, dimethyl phthalate, di(2-ethylhexyl) phthalate, diisobutyl phthalate, diisooctyl phthalate, etc. Further studies reveal that PAEs can be biosynthesized by at least several algae. PAEs are reported to possess allelopathic, antimicrobial, insecticidal, and other biological activities, which might enhance the competitiveness of plants, algae, and microorganisms to better accommodate biotic and abiotic stress. These findings suggest that PAEs should not be treated solely as a “human-made pollutant” simply because they have been extensively synthesized and utilized; on the other hand, synthesized PAEs entering the ecosystem might disrupt the metabolic process of certain plant, algal, and microbial communities. Therefore, further studies are required to elucidate the relevant mechanisms and ecological consequences.

## 1. Introduction

Phthalic acid esters (PAEs) are common plasticizers added to polymeric materials to improve their flexibility and workability [1]. PAEs have been widely used in numerous consumer products, including cosmetics, food packaging, building materials, medical supplies, home furnishings, etc., due to their characteristic properties, such as their good insulation, high strength, excellent corrosion resistance, low cost, and ease of fabrication [2,3,4]. The current global production of PAEs is estimated at 300 million tons, and it is expected to reach 500 million tons by 2050, most of which will be single-use products [5]. Moreover, China has become the world’s largest producer, consumer, and importer of plasticizers, accounting for nearly 42% of the world’s consumption in 2017 [6]. As one of the most abundantly produced phthalates, di(2-ethylhexyl) phthalate accounts for one-third and 80% of the phthalates made in the European Union and China, respectively [7]. With such extensive application of phthalate-containing products, PAEs have attracted increasing attention as environmental and biomedical pollutants, which may invisibly enter the human body through airborne transmission, skin contact, and food chain transmission, constituting potential health and ecological system threats [8]. In fact, a number of studies have been carried out to investigate the toxicity of PAEs on human beings and/or animals. Epidemiologic studies found that early phthalates exposure could induce significant neuro-developmental damage [9]. Some PAEs have been proven to possess reproductive and developmental toxicities to animals and are suspected of causing endocrine-disrupting effects to humans [10,11,12]. PAEs were also harmful to aquatic organisms. Di-*n*-butyl, diethyl phthalate, and their mixture were found to effectively activate zebrafish embryos’ antioxidant system and lead to immunotoxicity and neurotoxicity [13,14]. Zhao et al. (2014) reported that di-*n*-butyl and di(2-ethylhexyl) phthalate disrupted the antioxidant system of carps, meanwhile combined exposure to these two compounds exacerbated this change [15].

Up to now, most of the published literature has focused on the detection methods, pollution distribution, and toxicological hazards of PAEs. However, the natural sources of various PAEs are rarely studied. The first report of phthalic acid as a natural substance was conducted by Schmid and Karrer (1945) [16] and, since then, more than 50 different derivatives of PAEs have been reported from different taxonomic groups, including bacteria, actinomycetes, fungi, fern, higher plants, and even animals [17]. What remains unclear, however, is that in many cases, it is rather complicated to determine whether these compounds come from synthesized materials that later cause contamination of the air, water, or soil, or whether they may be produced by the plants and microorganisms themselves. The objective of this review is to summarize the plant and microorganism origin of PAEs so as to better understand their possible sources: Are they synthesized chemicals, or are they naturally occurring secondary metabolites?

## 2. Physicochemical Properties and Applications of PAEs

Phthalic acid esters (dialkyl or alkyl aryl esters of 1,2-benzenedicarboxylic acid), usually called PAEs, phthalate esters, or just phthalates, are a group of important derivatives of phthalic acids which are synthesized from phthalic anhydride and specific alcohols by Fischer esterification [18,19]. PAEs based on hydrogen bond and van der Waals force interconnection are hydrophobic compounds with log *K*_ow_ ranging from 1.6 to 12 [20]. Most of the phthalate esters are colorless liquids with a low volatility, high boiling point, and poor solubility in water, but they are soluble in organic solvents and oils [8]. These esters’ general chemical structure consists of a rigid planar aromatic ring and two malleable nonlinear fatty side chains. The two side-chain groups can be the same or not, and there are approximately 30 types of different side chains, ranging from dimethyl phthalate to tridecyl ester [21]. Due to phthalate esters being widespread in the biosphere and potential hazards in relation to ecosystem functioning and public health, six PAEs have been listed as priority pollutants by the United States Environmental Protection Agency and the European Union [20,22], including dimethyl phthalate, diethyl phthalate, di-*n*-butyl phthalate, butyl benzyl phthalate, di(2-ethylhexyl) phthalate, and di-*n*-octyl phthalate. These phthalate esters’ physicochemical properties and common applications are summarized in Table 1 and Figure 1.

PAEs are a class of lipophilic chemicals widely used in the plastics manufacturing industries as plasticizers and additives to improve the mechanical extensibility and flexibility of various products, such as plastics, paints, and synthetic fibers [23]. Phthalates of lower molecular weight, such as dimethyl phthalate, diethyl phthalate, and di-*n*-butyl phthalate, are widely used in cosmetics and personal care products; dimethyl phthalate and diethyl phthalate allow perfume fragrances to evaporate more slowly, making the scent linger longer, and a small amount of di-*n*-butyl phthalate can make nail polish chip-resistant. Di-*n*-butyl phthalate is also used in cellulose esters, printing inks, latex adhesives, and insect repellents [11,24].

Higher phthalate molecules, such as di(2-ethylhexyl) phthalate, diisononyl phthalate, and butyl benzyl phthalate, have a wide range of applications as plasticizers in the polymer industry to improve flexibility, workability, and general handling properties, and about 80% of PAEs are used for this purpose [20,25]. The stability, fluidity and low volatility of these compounds make them very suitable for manufacturing PVC and other resins, such as polyvinyl acetates and polyurethanes [26]. One of the most widespread phthalate plasticizers, di(2-ethylhexyl) phthalate, has several useful applications in numerous consumer products, commodities, and building materials [27]. Diisononyl phthalate is commonly used in garden hoses, pool liners, flooring tiles, tarps, and toys. Additionally, butyl benzyl phthalate, as a component of materials, is extensively used in vinyl flooring, synthetic leather, inks, and adhesives [19]. Phthalates are not covalently bound to the polymer matrix, rather they usually remain present as a freely mobile and leachable phase; therefore, they can be lost from soft plastic over time and released to the environment during production and manufacture. Not surprisingly, phthalates can often be found in freshwater lakes and oceans [28,29], urban and suburban soil [30,31], the atmosphere [32,33], and sediments [34,35]. Bu et al. (2020) [36] summarized the concentrations of six representative phthalates from published papers in the last twenty years (2000–2019) to analyze the pollution characteristics of phthalates worldwide and found that their mean concentration in settled dust was 500.02 μg/g in North America, 580.12 μg/g in Europe, and 945.45 μg/g in Asia, with DEHP being the most predominant phthalate, with mean and median values of 615.78 μg/g and 394.03 μg/g, respectively; the mean concentration of six representative phthalates in indoor air was 598.14 ng/m^3^ in North America, 823.98 ng/m^3^ in Europe, and 1710.26 ng/m^3^ in Asia. In another study, Hu et al. (2020) [37] detected 8 PAEs in 67 sediment samples collected from Hangzhou Bay, Taizhou Bay, and Wenzhou Bay in China; the total concentrations of detected PAEs were in the range of 654–2603 ng/g, with di(2-ethylhexyl) phthalate being the predominant PAE (mean 663 ng/g, accounting for a mean of 52% of total PAEs).

**Table 1 toxins-13-00495-t001:** Physicochemical properties and application of six PAEs listed as priority pollutants.

PAEs	Molecular Formula	Molecular Weight	CAS Registration Number	Specific Gravity (20 °C)	Water Solubility (mg/L)	log *K_ow_*	Melting Point (°C)	Application	References
Dimethyl phthalate	C_10_H_10_O_4_	194.18	131-11-3	1.19	4000	1.47	5.5	Insect repellent, personal care products, etc.	[12]
Diethyl phthalate	C_12_H_14_O_4_	222.24	84-66-2	1.12	1000	2.38	–40	Personal care products, plasticizers, cosmetics, etc.	[38]
Di-*n*-butyl phthalate	C_14_H_38_O_4_	278.35	84-74-2	1.05	11.2	3.74	–35	PVC plastics, explosive materials, nail paints, etc.	[39]
Butyl benzyl phthalate	C_19_H_20_O_4_	302.39	85-68-7	1.11	2.7	4.59	–35	Rapping materials, food conveyor belts, artificial letter, traffic cones, etc.	[40]
Di(2-ethylhexyl) phthalate	C_24_H_38_O_4_	390.62	117-81-7	0.99	0.003	7.5	–40	Medical devices, food packaging, building products, children’s products, etc.	[41]
Di-*n*-octyl phthalate	C_24_H_38_O_4_	390.62	117-84-0	0.99	0.0005	8.06	–25	Conveyor belts, pool liners, garden hoses, etc.	[22]

## 3. Natural Existence of PAEs in Living Organisms

### 3.1. PAEs from Plant Sources

Literature surveys revealed that PAEs were previously detected in different parts (stems, leaves, flowers, fruits, roots, and seeds) of 60 plant species that belong to 38 families, as well as in various algae, such as *Gracilaria lemaneiformis*, *Chaetomorpha basiretorsa*, and *Cladophora fracta* (Figure 2). PAEs were often found in the following families: Lamiaceae (seven species, accounting for 11.7% of the total), Rosaceae (four species, accounting for 6.7%), Solanaceae (four species, accounting for 6.7%), Liliaceae (three species, accounting for 5%), and Asteraceaeare (three species, accounting for 5%) (Table 2), which represented 35% of the total species, with di-*n*-butyl phthalate, diisobutyl phthalate, and di(2-ethylhexyl) phthalate being the most frequently detected PAEs.

PAEs have been detected via GC/MS in the organic extracts of certain plant species, with their percentages varying from 1.0% to 32.0% (Table 2). For instance, di-*n*-butyl phthalate was found in the extracts of *Brassica oleracea* (32.0%), *Ixora amplexicaulis* (15.0%), *Gossypium hirsutum* (7.9%), and *Zea mays* (7.0%) [42,43,44,45]. Di-*n*-octyl phthalate was identified to be abundant in the extracts of *Prunella vulgaris* (29.9%), *Jatropha curcas* (21.6%), and *Photinia parvifolia* (10.1%) [46,47,48]. Other PAEs, such as diethyl phthalate, isobutyl octyl phthalate, etc., were also reported in different plant extracts. Noteworthily, some of the detected PAEs are not commonly used in industry, implying that they might originate from biosynthesis rather than from contaminated soil or air.

Most PAEs were found in plant-derived essential oils (EOs). EOs can be synthesized by all plant organs (flowers, buds, seeds, leaves, twigs, bark, herbs, wood, fruits, and roots), which can be extracted using traditional hydrodistillation, organic solvent-steam distillation, headspace solid-phase microextraction (HS-SPME), and supercritical CO_2_ fluid extraction (CO_2_-SFE) procedures [49]. EOs not only play an important role in many physiological and biochemical reactions, but are also widely utilized in pharmaceutical, sanitary, cosmetic, agricultural, and food industries [50,51]. PAEs are constantly being identified in different varieties of Eos. Twenty-six plants have been reported to contain PAEs, with di-*n*-butyl phthalate being the most abundant constituent, which has been found in eighteen species, with the percentage ranging from 1.5% to 87.2% (Table 2). Species that are rich in di-*n*-butyl phthalate include *Radix pseudostellaria* (87.2%), *Clerodendrum inerme* (59.3%), *Pyrola rotundifolia* (40.5%), *Osmanthus fragrans* (15.1%), and *Alocasia macrorrhiza* (14.4%) [52,53,54,55,56]. Di(2-ethylhexyl) phthalate is also a common component detected in Eos; for instance, it is found in the Eos produced by *Cirsium japonicum* (30.8%), *Pyrus ussriensis* (29.4%), *Ziziphus mauritiana* (18.0%), and *Clerodendrum inerme* (17.3%) [56,57,58,59].

Some PAEs were found in the litter and root exudates of plants, which are actually considered the primary inputs of allelochemicals to the external environment that affect neighboring plants’ growth [60]. At least in part, allelopathy helps explain the mechanism of the establishment of dominance of certain plant species, including invasive alien species; allelopathy also provides a theoretical basis for revealing the mechanism of crop intercropping and rotation obstacles in agricultural production [60,61]. In fact, some PAEs, such as di-*n*-octyl phthalate, have been confirmed to be active allelochemicals [62]. Di-*n*-butyl phthalate and diisobutyl phthalate are the most frequently identified PAEs in root exudates of plants such as *Solanum lycopersicum*, *Capsicum annuum*, *Z**. mays*, *Solanum melongena*, etc. (Table 2). Cheng and Xu (2012) [63] collected root exudates of *Lilium brownii*, which revealed that phthalate acid esters, such as diisooctyl phthalate (52.1%) and di(2-ethylhexyl) phthalate (41.0%), were dominant. Zhou et al. (2010) [64] studied the root exudates of grafted eggplants using the root soaking method, which led to the identification of di-*n*-butyl phthalate (13.6%), diisobutyl phthalate (1.9%), and diisononyl phthalate (0.8%). GC–MS analysis showed that there were eleven organic compounds in the methanol extract of root exudates of *Allium fistulosum*, including derivatives of phthalate ester, such as diisooctyl phthalate (52.1%) and di(2-ethylhexyl) phthalate (41.0%).

Although the GC/MS procedure is effective in detecting PAEs, it has its limitations. In some studies, calculation of the retention indices (RI) was ignored; thus, the accuracy of the identification of PAEs was reduced. Traditionally, preparative chromatographic purification of secondary metabolites produced by plants includes the application of silica gel column chromatography, sephadex LH−20 gel column chromatography, semi-preparative HPLC, preparative TLC, etc. During this process, PAEs such as di-*n*-butyl phthalate, diisobutyl phthalate, etc., were purified from different plant species (Table 2). Liu et al. (2011) [65] isolated di-*n*-butyl phthalate and diisobutyl phthalate from the leaves and stems of *Toona ciliata*. Shi et al. (2005) [66] obtained di-*n*-butyl phthalate and diisobutyl phthalate from *C. basiretorsa* for the first time by spectroscopic methods. As secondary metabolites, di-*n*-butyl phthalate and diisobutyl phthalate were also isolated from the whole plants of *C. fracta* [67], the root of *Croton lachynocarpus* [68], and the fruits of *Pyrus bretschneideri* [69]. Consequently, PAEs identified and purified in plant materials illustrate that the plants could synthesize them to some extent.

**Table 2 toxins-13-00495-t002:** PAEs detected in plant materials.

Family	Identified from	Origin	Type of PAEs	Relative Content of PAEs (%) *	References
Acanthaceae	*Avicennia marina*	Fruits	Diethyl phthalate	1.2	[70]
			Dimethyl phthalate	0.6	
			Methyl nonyl phthalate	0.4	
	*Asystasia gangetica*	Aerial Parts	Diisobutyl phthalate	6.1	[71]
			Bis-Decyloctyl phthalate	5.7	
			Bis-Diundecyl phthalate	5.7	
			Bis-Decylhexyl phthalate	4.2	
			Bis-isodecylhexyl phthalate	4.1	
			Diheptyl phthalate	3.6	
			Bis-Didecyl phthalate	2.6	
			Bis-Heptyloctyl phthalate	2.4	
			Di-*n*-butyl phthalate	2.3	
			Di(2-ethylhexyl) phthalate	1.5	
			Bis-7-Methy loctyl phthalate	1.0	
Araceae	*Alocasia macrorrhiza*	Whole Plants	Bis (2-isobutyl) phthalate	32.5	[55]
			Di-n-butyl phthalate	14.4	
Asteraceae	*Ageratina adenophora*	Leaves, Shoots	Di(2-ethylhexyl) phthalate	N/A **	[72]
			Di-*n*-butyl phthalate		
	*Cirsium japonicum*	Whole Plants	Di(2-ethylhexyl) phthalate	30.8	[57]
			Diisooctyl phthalate	16.6	
			Mono (2-ethylhexyl) phthalate	16.0	
			Diisobutyl phthalate	1.1	
			Butyloctyl phthalate	0.7	
			Di-*n*-octyl phthalate	0.1	
	*Chrysanthemum indicum*	Leaves, Stems	Diethyl phthalate	N/A	[73]
Apiaceae	*Angelica sinensis*	Roots	Di-*n*-butyl phthalate	N/A	[74]
			Di(2-ethylhexyl) phthalate		
			Bis (2-methylpropyl) phthalate		
Brassicaceae	*Brassica oleracea*	Stalks	Di-*n*-butyl phthalate	32.0	[42]
			Diisooctyl phthalate	18.5	
			Diisobutyl phthalate	3.4	
			Diethyl phthalate	1.3	
Chenopodiaceae	*Beta vulgaris*	Root Exudates	Di-*n*-butyl phthalate	47.2	[75]
			Diisobutyl phthalate	8.6	
Campanulaceae	*Campanula colorata*	Whole Plants	Butyloctyl phthalate	10.2	[76]
			Di-*n*-butyl phthalate	7.4	
			Diisooctyl phthalate	0.6	
Calycanthaceae	*Chimonanthus praecox*	Flowers	Di-*n*-butyl phthalate	4.5	[77]
Cladophoraceae	*Cladophora fracta*	Whole Plants	Diisobutyl phthalate	N/A	[67]
			Di-*n*-butyl phthalate		
	*Chaetomorpha basiretorsa*	Whole Plants	Di-*n*-butyl phthalate	N/A	[66]
			Diisobutyl phthalate		
Cyperaceae	*Fimbristylis miliacea*	Whole Plants	Di-*n*-octyl phthalate	N/A	[62]
Crassulaceae	*Hylotelephium erythrostictum*	Flowers	Di-*n*-butyl phthalate	1.2	[78]
Convolvulaceae	*Ipomoea carnea*	Whole Plants	Di-*n*-butyl phthalate	N/A	[79]
Caryophyllaceae	*Radix pseudostellariae*	Whole Plants	Di-*n*-butyl phthalate	87.2	[52]
			Ditridecyl phthalate	0.7	
Euphorbiaceae	*Croton lachynocarpus*	Roots	Di-*n*-butyl phthalate	N/A	[68]
			Diisobutyl phthalate		
			Butyl isobutyl phthalate		
	*Jatropha curcas*	Leaves	Di-*n*-octyl phthalate	21.6	[46]
Ericaceae	*Pyrola rotundifolia*	Whole Plants	Di-*n*-butyl phthalate	40.5	[53]
	*Rhododendron calophytum*	Flowers	Di-*n*-butyl phthalate	4.9	[80]
			Diisobutyl phthalate	1.4	
Fabaceae	*Dalbergia odorifera*	Flowers	Di-*n*-butyl phthalate	14.0	[81]
			Diisooctyl phthalate	4.4	
	*Medicago sativa*	Root Exudates	Di-*n*-butyl phthalate	10.7	[82]
Gracilariaceae	*Gracilaria lemaneiformis*	Whole Plants	Butyl isobutyl phthalate	N/A	[83]
Gesneriaceae	*Lysionotus pauciflorus*	Whole Plants	Diisobutyl phthalate	2.7	[84]
Hypericaceae	*Hypericum scabrum*	Seeds, Leaves	Di(2-ethylhexyl) phthalate	5.8	[85]
Liliaceae	*Allium fistulosum*	Root Exudates	Diisooctyl phthalate	11.4	[86]
			Di-*n*-butyl phthalate	4.7	
			Diethyl phthalate	3.2	
			Dimethyl phthalate	0.9	
			Diisobutyl phthalate	0.7	
			Butyl methyl phthalate	0.6	
	*Lilium brownii*	Root Exudates	Diisooctyl phthalate	52.1	[63]
			Di(2-ethylhexyl) phthalate	41.0	
			Methyl 2-ethylhexyl phthalate	0.9	
			2-ethyl hexyl butyl phthalate	0.8	
			Di-*n*-butyl phthalate	0.3	
	*Paris polyphylla*	Roots	Isobutyl-3-pentenyl phthalate	24.7	[87]
			Butyl-2-isobutyl phthalate	5.5	
			Di(2-ethylhexyl) phthalate	4.2	
Lamiaceae	*Clerodendrum inerme*	Leaves	Di-*n*-butyl phthalate	59.3	[56]
			Di(2-ethylhexyl) phthalate	17.3	
	*Melissa officinalis*	Aerial Parts	Diisobutyl phthalate	2.5	[88]
			Di-*n*-butyl phthalate	1.4	
	*Ocimum obovatum*	Leaves	2-ethylhexyl undecyl phthalate	5.3	[89]
			Di-*n*-butyl phthalate	4.5	
	*Phlomis umbrosa*	Flowers	Diisobutyl phthalate	13.4	[90]
			Di-*n*-butyl phthalate	1.5	
			Butyl isobutyl phthalate	0.4	
	*Prunella vulgaris*	Whole Plants	Di-*n*-octyl phthalate	29.9	[47]
			Diethyl phthalate	2.5	
	*Phlomis medicinalis*	Roots	Butyl isobutyl phthalate	N/A	[91]
	*Scutellaria barbata*	Whole Plants	Di-*n*-butyl phthalate	8. 3	[92]
			Diisobutyl phthalate	3. 6	
Malvaceae	*Gossypium hirsutum*	Stalks	Di-*n*-butyl phthalate	7.9	[45]
Myricaceae	*Myricarubra sieb*	Fruits	Phthalic acid, hex-3-yl isobutyl ester	9.7	[93]
			Diisooctyl phthalate	4.2	
			Di-*n*-butyl phthalate	2.0	
			Dimethyl phthalate	0.8	
Meliaceae	*Toona ciliata*	Leaves, Stems	Diisobutyl phthalate	N/A	[65]
			Di-*n*-butyl phthalate		
Orchidaceae	*Cymbidium sinense*	Flowers	Diisobutyl phthalate	12.5	[94]
Oleaceae	*Osmanthus fragrans*	Flowers	Mono (2-ethylhexyl) phthalate	26.5	[54]
			Bis (2-methylpropyl) phthalate	21.9	
			Di-*n*-butyl phthalate	15.1	
			Diethyl phthalate	2.1	
Pontederiaceae	*Eichhornia crassipes*	Whole Plants	Di-*n*-octyl phthalate	N/A	[95]
			Diisooctyl phthalate		
			Mono (2-ethylhexyl) phthalate		
			Methyl dioctyl phthalate		
Polygonaceae	*Polygonum amplexicaule*	Roots	Diisobutyl phthalate	N/A	[96]
Poaceae	*Zea mays*	Straws	Di-*n*-butyl phthalate	7.0	[44]
			2-Methyl-pentyl-isobutyl phthalate dibutyl	6.4	
Rosaceae	*Malus prunifolia*	Root Exudates	Phthalate derivates	52.5	[97]
	*Pyrus bretschneideri*	Seeds	Di-*n*-butyl phthalate	N/A	[69]
			Diisobutyl phthalate		
	*Pyrus ussriensis*	Fruits	Di(2-ethylhexyl) phthalate	29.4	[58]
	*Photinia parvifolia*	Fruits	Di-*n*-octyl phthalate	10.1	[48]
Rubiaceae	*Paederia scandens*	Whole Plants	Di-*n*-butyl phthalate	5.0	[98]
			Dimethyl phthalate	3. 7	
			Diisobutyl phthalate	3.2	
			Di-*n*-octyl phthalate	2.9	
	*Ixora amplexicaulis*	Branches, Leaves	Di-*n*-butyl phthalate	15.0	[43]
Rhamnaceae	*Ziziphus mauritiana*	Fruits	Di(2-ethylhexyl) phthalate	18.0	[59]
			Di-*n*-butyl phthalate	12.3	
Solanaceae	*Capsicum annuum*	Leaves and Root Exudates	Di-*n*-butyl phthalate	41.5	[99]
			Butyl cyclohexane phthalate	15.6	
			Butyl isobutyl phthalate	13.1	
			Ditert butyl phthalate	10.1	
	*Nicotiana tabacum*	Root Exudates	3-hexyl isobutyl phthalate	4.8	[100]
			Diisobutyl phthalate	2.9	
	*Solanum lycopersicum*	Root Exudates	Di-*n*-butyl phthalate	5.8	[101]
			Dimethyl phthalate	2.1	
			Diisooctyl phthalate	1.7	
			Diisobutyl phthalate	0.4	
	*Solanum melongena*	Root Exudates	Di-*n*-butyl phthalate	13.6	[64]
			Diisobutyl phthalate	1.9	
			Diisononyl phthalate	0.8	
Saxifragaceae	*Saxifraga stolonfera*	Whole Plants	Butyloctyl phthalate	5.5	[102]
Sargassaceae	*Nizamuddinia zanardinii*	Whole Plants	Di-*n*-butyl phthalate	5.1	[103]
			Diethyl phthalate	0.7	
Sapindaceae	*Nephelium lappaceum*	Peels	Isobutyl octyl phthalate	16.5	[104]
			Diisooctyl phthalate	8.9	
Salviniaceae	*Salvinia natans*	Whole Plants	Mono (2-ethylhexyl) phthalate	29.3	[105]
			Di-*n*-butyl phthalate	1.0	
Thymelaeaceae	*Stellera chamaejasme*	Root Exudates	2-Ethyl hexyl phthalate	18.7	[106]
			Di-*n*-butyl phthalate	4.6	
			Diisobutyl phthalate	0.2	

* Relative Content of PAEs (%) detected via GC/MS; ** N/A: Not applicable.

### 3.2. PAEs Identified and Purified from Microorganisms

Phthalate compounds as bioactive natural products can be produced not only by plants, but also by bacteria and fungi (Table 3). Keire et al. (2001) [38] reported the first known example of diethyl phthalate produced by a bacterium, *Helicobacter pylori*, which represents a new class of immune-modulatory agent. Aboobaker et al. (2019) [107] isolated di-*n*-butyl phthalate as the major bioactive compound from the endophytic fungi, *Pelargonium sidoides*, which exhibits a significant inhibitory effect on Gram-positive bacteria (*Staphylococcus aureus* and *Enterococcus faecalis*) and Gram-negative bacteria (*Escherichia coli* and *Pseudomonas aeruginosa*). Rajamanikyam et al. (2017) [108] purified two PAEs, di(2-ethylhexyl) phthalate and di-*n*-butyl phthalate, from *Brevibacterium mcbrellneri*, both of which were isolated for the first time from the bacteria. Di-*n*-butyl phthalate was isolated from *Streptomyces melanosporofaciens* as an effective inhibitor of α-glucosidase, which could provide useful reference information for the design of new effective inhibitors of glycosidase [109]. Furthermore, di(2-ethylhexyl) phthalate was isolated from *Streptomyces bangladeshensis* [110] and *Penicillium olsonii* [111]. Therefore, it is expected that PAEs can be characterized in various microorganisms, although their sources remain unclear.

## 4. Biological Activities of PAEs

### 4.1. Allelopathic/Phytotoxic Activity

Allelopathy refers to any direct or indirect harmful or beneficial effect exerted by one plant on another through the production of chemical compounds that are released into the environment. In some cases, allelopathy is suspected to contribute to the establishment of dominance of certain plant species, including some invasive alien species. Due to the phytotoxic property of allelochemicals, they are often considered valuable candidates for environmentally friendly bioherbicies [113,114]. Di-*n*-octyl phthalate isolated from *Fimbristylis miliacea* can remarkably inhibit the seed germination of tested weed species *Ludwigia hysopifolia, Echinochloa colonum, Cyperus iria*, and *Paspalam digitatum* [62]. Zhu et al. (2014) [72] isolated two allelochemicals, di(2-ethylhexyl) phthalate and di-*n*-butyl phthalate, from the root exudates of the invasive plant, *Ageratina adenophora*. In a bioassay, di-*n*-butyl phthalate was found to possess a significant inhibitory effect on seed germination and seedling growth of *A. adenophora*. Meanwhile, these two compounds significantly increased the superoxide dismutase (SOD) activity of *A. adenophora*’s leaves and caused lipid peroxidation and cell membrane damage. Xuan et al. (2006) [115] identified the derivatives of phthalic acids from root exudates of *Echinochloa crusgalli* and found that diethyl phthalate strongly affects the seedling growth of alfalfa, Indian jointvetch, lettuce, monochorea, and sesame. Huang et al. (2017) [116] analyzed the extracts of aerial parts plants, root exudates, and plant rhizosphere soil of *Chrysanthemum indicum* to determine the effect of the allelochemical diethyl phthalate, and the results show that it has a noticeable impact on promoting the fresh weight of lettuce, as well as the root growth of lettuce and rape. Shanab et al. (2010) [95] extracted four phthalate derivatives from *Eichhornia crassipes*, including di-*n*-octyl phthalate, mono (2-ethylhexyl) phthalate, methyl dioctyl phthalate, and diisooctyl phthalateis, which possess strong inhibitory effects on *Chlorella vulgaris*.

Physiological studies have indicated that PAEs can influence enzyme activity, which might be at least one of their phytotoxicity mechanisms. Deng et al. (2017) [117] revealed that as the concentration of PAEs secreted by tobacco roots increased, the rate of production of superoxide anion radicals, the concentration of malondialdehyde, and the activity of peroxidase and SOD in tobacco root increased significantly. A series of changes could reduce the root system’s antioxidant properties and cause oxidative damage to the apical cell membrane system, thereby affecting root absorption and ultimately showing autotoxicity. Dong et al. (2016) [67] extracted diisobutyl phthalate and di-*n*-butyl phthalate from the ethyl acetate extract of *C. fracta*, both of which show a strong inhibitory effect on the growth of *Heterosigma akashiwo* and *Gymnodinium breve*, which may be related to the production of reactive oxygen species (ROS) induced by diisobutyl phthalate and di-*n*-butyl phthalate in algal cells. Excessive ROS inhibits the activities of catalase and SOD, leading to lipid oxidation and the destruction of algae cell membranes.

### 4.2. Antimicrobial Activity

Natural products, including secondary metabolites produced by plants and microorganisms, have long been studied for their antimicrobial activity in the search for eco-friendly substitutes for synthesized chemicals [118]. Di(2-ethylhexyl) phthalate and di-*n*-butyl phthalate isolated from *B. mcbrellneri* show broad-spectrum antibacterial activity [108]. Di(2-ethylhexyl) phthalate can inhibit the growth of gram-positive (*S. epidermidis*, MIC of 9.37 µg/mL; *S. aureus*, MIC of 18.75 µg/mL) and gram-negative bacteria (*E. coli*, MIC of 37.5 µg/mL; *P. aeruginosa* and *Klebsiella pneumoniae*, MIC at 75 µg/mL for both). Di-*n*-butyl phthalate also inhibits the growth of gram-positive (*Bacillus subtilis* and *S. epidermidis*, MIC at 18.75 µg/mL for both) as well as gram-negative bacteria (*E. coli* and *P. aeroginosa*, MIC at 37.5 µg/mL for both). Di(2-ethylhexyl) phthalate isolated from the flowers of *Calotropis gigantean* exerts antimicrobial activity against *B. subtilis* with a MIC of 32 µg/mL [119]. There are also reports on the antimicrobial activity of di-*n*-butyl phthalate isolated from *Streptomyces albidoflavus* showing a MIC for *E. coli* of 53 µg/mL, with *B. subtilis* at 84 µg/mL [112]. Four phthalate derivatives isolated from *E. crassipes* also exert significant antibacterial activity against gram-positive bacteria (*B. subtilis* and *Streptococcus faecalis)* and gram-negative bacteria *E. coli*, and antifungal activity against *Candida albicans* [95]. In another study, El-Mehalawy et al. (2008) [120] found that di(2-ethylhexyl) phthalate could be produced by certain bacteria, including *Tsukamurella inchonensis, Corynebacterium nitrilophilus*, and *Cellulosimicrbium cellulans*, and di(2-ethylhexyl) phthalate has the function to inhibit fungal spore germination, cell membrane growth, and the production of total lipids and total protein. Li et al. (2021) [121] isolated di-*n*-butyl phthalate from a new marine *Streptomyces* sp. and found this compound significantly inhibited spore germination and mycelial growth of *Colletotrichum fragariae*. In addition to this, an obvious decrease was detected in sugar and protein contents of *C. fragariae* mycelia. Other studies have shown similar results. For instance, di-*n*-butyl phthalate was reported to inhibit spore germination and mycelium growth of *Colletotrichum gloeosporioides*, *Colletotrichum musae*, and *Gaeumannomyces graminis* [122,123,124].

Janu and Jayanthy (2014) found that diethyl phthalate derived from the fungus *Aspergillus* sp. increased the superoxide production and exerted ROS generated oxidative stress in the cytoplasm of bacterial cells, which eventually led to cell death [125]. In addition, diethyl phthalate with antimicrobial properties was reported for its ability to interfere with quorum sensing mediated virulence factors and biofilm formation in *Pseudomonas aeruginosa* [126,127]. Another study demonstrated that dimethyl phthalate (concentration ranged from 20 to 40 mg/L) greatly inhibited the growth and glucose utilization of *Pseudomonas fluorescens*, meanwhile the surface hydrophobicity and membrane permeability of *P. fluorescens* were also increased. Dimethyl phthalate could lead to deformation of the cell membrane and misopening of membrane channels. Additionally, RNA-Seq and RT-qPCR results revealed that the expression of some genes in *P. fluorescens* were altered, including the genes involved in energy metabolism, ATP-binding cassette transporting, and two-component systems by dimethyl phthalate [128].

### 4.3. Insecticidal Activity

In addition to their phytotoxic and antimicrobial activity, PAEs were also found to be insecticidal; attributed to inhibition of acetylcholinesterase enzyme activity, they possess significant mosquito larvicidal activity. Therefore, some phthalates, such as synthetic diethyl phthalate and dimethyl phthalate, have been used as active ingredients in insect repellents [129,130]. Previously, Adsul et al. (2012) [79] isolated di-*n*-butyl phthalate from the leaf extract of *Ipomoea carnea* via column chromatography, and this compound ws found to be lethal to the fourth instar larvae of *Aedes aegypti* and *Culex quinquefasciatus*, with the lethal concentrations of LC_50_ being 81.43 and 109.64 ppm, respectively. Di-*n*-butyl phthalate and di(2-ethylhexyl) phthalate were isolated from the bacterium *B. mcbrellneri;* because of their significant acetylcholinesterase inhibitory activity, they are also active against the fourth instar of *A. aegypti* after 24 h of exposure [108]. On the other hand, various PAEs were also constantly reported as possessing other biological activities, such as anti-inflammatory, antiviral, anti-tumor, antidiabetic activity, etc., indicating their valuable potential to be explored further in capacities other than plasticizers [17] (Figure 3).

## 5. Conclusions and Perspectives

PAEs have attracted attention due to their ubiquitous presence in environmental media. Many phthalates have been reported in nature, such as sediments, natural water, soil, aquatic organisms, etc. [17,20,30,34]. On the other hand, as phthalates exist in many laboratory products, such as instruments, reagents, solvents, and consumables, source detection also poses analytical challenges [131,132,133]. PAEs in the environment are mainly derived from chemical syntheses that are applied in building materials, care products, medical equipment, and children’s toys, etc., which are convenient for human production and life [19,27,132]. However, the published literature also indicated that PAEs can be synthesized naturally, and they might serve as biologically active substances to enhance competitiveness. The isotope labeling approach has demonstrated that PAEs can be biosynthesized by several algae, possibly via the shikimic acid pathway [134,135,136]. Ecologically, PAEs with allelopathic activity might facilitate the establishment of the dominance of plants or algae that are capable of producing them. In addition, phthalate esters’ insecticidal activity protects plants from being consumed by insects [17,129,137], not to mention the fact that the antimicrobial activity of PAEs may reduce the damage caused by pathogenic fungi and bacteria [17,107,108,118,119].

Studies have shown that some algae can also synthesize phthalates to defend against biotic and abiotic factors. Babu and Wu (2010) [138] highlighted that some freshwater algae and cyanobacteria species are capable of producing di-*n*-butyl phthalate and mono (2-ethylhexyl) phthalate. These phthalates may be released into the environment under pressure, affecting the aquatic ecosystem. The above conclusion is in good agreement with Chen’s results, who reported on the de novo synthesis of di(2-ethylhexyl) phthalate and di-*n*-butyl phthalate in a marine alga. In algal cells, biosynthesized PAEs are presumably stored in cell membranes to maintain the flexibility of algal cells [134]. These findings suggest that the production of phthalates may be a common phenomenon on both land and in the sea. Meanwhile, some PAEs identified in the root exudates of various crops could effectively reduce soil-borne diseases, improve soil properties, and promote plant growth [64,101,139]. Nevertheless, the biosynthetic pathways of these secondary metabolites are highly complex and are the result of the combined actions of biotic and abiotic stressors, which are worthy of in-depth study by phytochemical researchers [134,138,140].

In conclusion, PAEs are widespread around us, not only from synthetic materials but also from living organisms, such as microbes, algae, plants, etc. Chemically synthesized PAEs have been widely applied in industry to improve the quality of various products. In contrast, naturally synthesized PAEs can potentially serve as allelochemicals, antibiotics, or insecticides to increase the adaptability of donor species. It is challenging to quantify the PAEs around us, and the amount of PAEs in the environment will continue to rise. We know that PAEs can be synthesized naturally, which implies that certain microorganisms are capable of degrading them, and their potential is worthy of further study to reduce PAEs’ contamination of the environment.

## Figures and Tables

**Figure 1 toxins-13-00495-f001:**
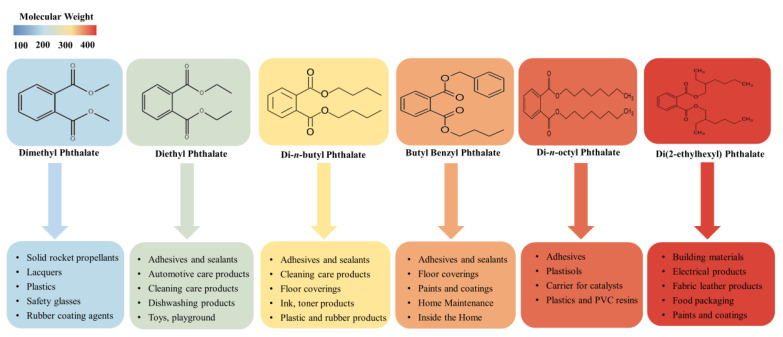
The application of six PAEs listed as priority pollutants.

**Figure 2 toxins-13-00495-f002:**
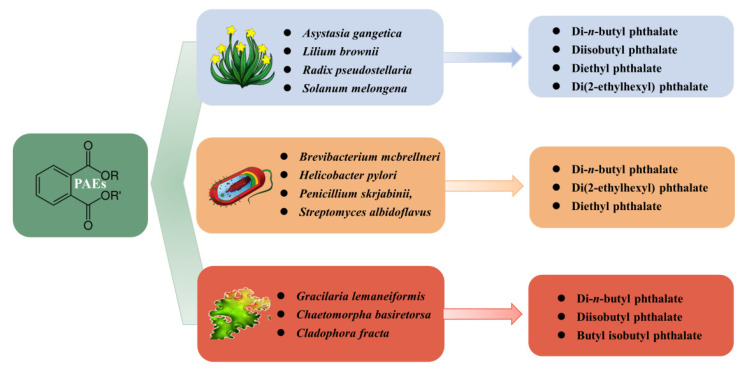
Natural existence of PAEs in living organisms.

**Figure 3 toxins-13-00495-f003:**
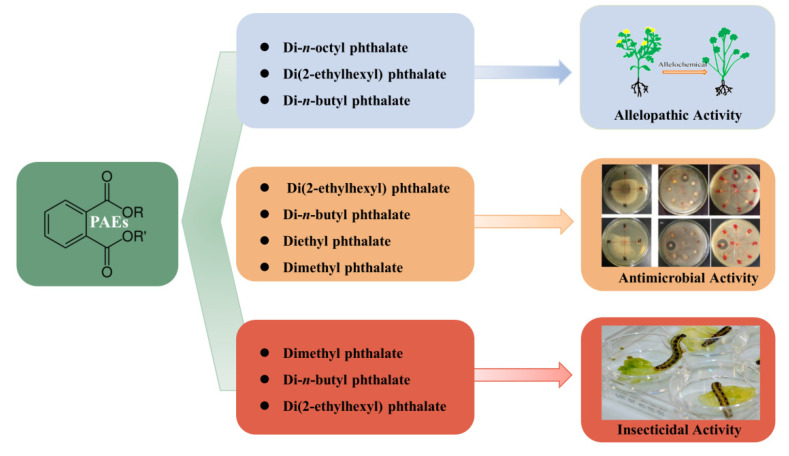
Biological activities of PAEs in living organisms.

**Table 3 toxins-13-00495-t003:** PAEs purified from microorganisms.

Category	Family	Species	Type of PAEs	References
Bacteria	Brevibacteriaceae	*Brevibacterium mcbrellneri*	Di(2-ethylhexyl) phthalate	[108]
			Di-*n*-butyl phthalate	
Fungi	Davidiellaceae	*Penicillium skrjabinii*	Di-*n*-butyl phthalate	[107]
Fungi	Davidiellaceae	*Penicillium olsonii*	Di(2-ethylhexyl) phthalate	[111]
Bacteria	Helicobacteraceae	*Helicobacter pylori*	Diethyl phthalate	[38]
Bacteria	Streptomycetaceae	*Streptomyces melanosporofaciens*	Di-*n*-butyl phthalate	[109]
Bacteria	Streptomycetaceae	*Streptomyces albidoflavus*	Di-*n*-butyl phthalate	[112]
Bacteria	Streptomycetaceae	*Streptomyces bangladeshensis*	Di(2-ethylhexyl) phthalate	[110]

## Data Availability

Not applicable.

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
