# Peer review of "Phthalic Acid Esters: Natural Sources and Biological Activities"

_toxins, 2021, doi:10.3390/toxins13070495_

Round 1

Reviewer 1 Report

Corrections have been performed as requested. The manuscript may be published in its current state under the editor's responsibility. 

Author Response

Reviewer #1 

-Corrections have been performed as requested. The manuscript may be published in its current state under the editor's responsibility.

Response: we greatly appreciate the reviewer’s efforts in improving the quality of our manuscript.

Reviewer 2 Report

This review deals with natural sources (organisms and plants) of phtalic acid esters (PAEs) and their biological activities. The manuscript is well organized and well written and some shortcomings can be easily amended by the Authors

  1. please, delete etc  at lines 117 and 126
  2. table 3- the reported microorganisms are not only bacteria, so, please add a first column (before Family) indicating bacteria and fungi (Fam Davidiellaceae). It is not correct to provide the family names only, for organisms belonging to different kingdoms.
  3. subheading "Antimicrobial activity" - can you please provide more information about the mechanisms of action of PAEs as antimicrobials? Lines 257 and following - Can you give some information about the fungal species whose spores where inhibited by PAEs?
  4. I was not able to find any information about the toxicity of PAEs for human beings and/or animals, including aquatic organisms, to evaluate the impact of these pollutants on the environment, can you add something about this aspect?

Reviewer 3 Report

The article entitled “Phthalic Acid Esters: Natural Sources and Biological Activities”, reports the  plant and microorganism origin of PAEs to better understand their possible sources: are they synthesized chemicals, or are they naturally occurring secondary metabolites. Studying the natural sources and biological activities of Phthalic Acid Esters, it is evident that a research field is of great interest, especially in the last few years. However, there some changes required, as reported below:

  • a graphical abstract would be very helpful for better understanding the article. Please insert one.
  • the article is very well worked and contains a lot of information but it would be better besides the tables to appear graphics for example: chapter 2. Physicochemical Properties and Applications of PAEs
  • in table 1 the references are missing please add them at each point

Round 2

Reviewer 2 Report

The revised version fully meets the reviewer's suggestions.